

# MOST: a modified MLST typing tool based on short read sequencing

Rediat Tewolde[1], Timothy Dallman[2], Ulf Schaefer[1], Carmen L. Sheppard[3], Philip Ashton[2], Bruno Pichon[4], Matthew Ellington[4], Craig Swift[2], Jonathan Green[1] and Anthony Underwood[1]

[1] Infectious Disease Informatics Unit, Public Health England, London, United Kingdom
[2] Gastrointestinal Bacteria Reference Unit, Public Health England, London, United Kingdom
[3] Respiratory and Vaccine Preventable Bacteria Reference Unit, Public Health England, London, United Kingdom
[4] Antimicrobial Resistance and Healthcare Associated Infection Unit, Public Health England, NIS, London, United Kingdom

Corresponding author
Rediat Tewolde,
rediat.tewolde@phe.gov.uk

## ABSTRACT

Multilocus sequence typing (MLST) is an effective method to describe bacterial populations. Conventionally, MLST involves Polymerase Chain Reaction (PCR) amplification of housekeeping genes followed by Sanger DNA sequencing. Public Health England (PHE) is in the process of replacing the conventional MLST methodology with a method based on short read sequence data derived from Whole Genome Sequencing (WGS). This paper reports the comparison of the reliability of MLST results derived from WGS data, comparing mapping and assembly-based approaches to conventional methods using 323 bacterial genomes of diverse species. The sensitivity of the two WGS based methods were further investigated with 26 mixed and 29 low coverage genomic data sets from *Salmonella enteridis* and *Streptococcus pneumoniae*. Of the 323 samples, 92.9% ($n = 300$), 97.5% ($n = 315$) and 99.7% ($n = 322$) full MLST profiles were derived by the conventional method, assembly- and mapping-based approaches, respectively. The concordance between samples that were typed by conventional (92.9%) and both WGS methods was 100%. From the 55 mixed and low coverage genomes, 89.1% ($n = 49$) and 67.3% ($n = 37$) full MLST profiles were derived from the mapping and assembly based approaches, respectively. In conclusion, deriving MLST from WGS data is more sensitive than the conventional method. When comparing WGS based methods, the mapping based approach was the most sensitive. In addition, the mapping based approach described here derives quality metrics, which are difficult to determine quantitatively using conventional and WGS-assembly based approaches.

## INTRODUCTION

The process of whole genome sequencing (WGS) has benefited from recent advances collectively known as next generation sequencing, allowing high throughput sequencing of bacterial genomes at low financial cost. This results in WGS becoming a viable alternative to some traditional typing methods for public health infectious disease surveillance.

MLST can be derived from WGS using *de novo* assembly/BLAST based (*Larsen et al., 2012*; *Jolley & Maiden, 2013*) and mapping based (*Inouye et al., 2012*; *Inouye et al., 2014*) approaches. *De novo* assembly/BLAST based approaches work by assembling short reads into longer contiguous sequences and then comparing these contigs to a reference allele database using BLAST to assign a MLST type. Mapping based approaches align short reads to reference (allele) sequences representing all alleles from MLST loci using mapping tools such as BWA (*Inouye et al., 2012*) or Bowtie2 (*Inouye et al., 2014*). Subsequently, SNP/INDELs are called using a variant-calling algorithm such as Samtools mpileup (*Li et al., 2009*) to determine the most likely allele at each locus. An allele is assigned if the reads have 100% coverage and 100% nucleotide identity to the locus alleles sequence without any INDELs. Mapping based approaches allow the calculation of metrics for each designated allele to assess the quality of the match (*Inouye et al., 2012*; *Inouye et al., 2014*).

Public Health England provides diagnostic, specialist and reference microbiology services to healthcare providers in England. Implementation of whole genome sequence (WGS) technology for public health microbiology requires quality controlled results that are at least as accurate as conventional 'gold standard' methods. In order to make an informed decision regarding the software that is most capable of accurately determining the MLST profile from WGS data, this paper systematically compared the performance of WGS-based MLST software to conventional methods using genomes from 323 samples. The software was evaluated based on the ability to: (a) Derive a full MLST profile, (b) demonstrate concordance to the MLST results derived from conventional sequencing and (c) assign quality metrics that allow results to be reported quantitatively.

# MATERIALS AND METHODS

## Isolates
Reference isolates and samples with mixed species assembly *in vitro* were prepared in order to compare the reliability of MLST results.

## Reference isolates
### Samples containing pure cultures of diverse bacteria
Isolates submitted to three different PHE reference laboratories, namely Gastrointestinal Bacteria Reference Unit (GBRU), Antimicrobial Resistance and Healthcare Associated Infection unit (AMRHAI) and Respiratory and Vaccine Preventable Bacteria Reference Unit (RVPBRU) were included for study.

The isolates were selected for WGS by each of the three units based on the following criteria:
A. RVPBRU receives submissions of all invasive *Streptococcus pneumoniae* from hospital laboratories in England and Wales for confirmation of species and for serotyping. From these, representatives of many different serotypes were selected.
B. AMRHAI receives isolates of *Staphylococcus aureus* from hospital laboratories in England & Wales for identification and molecular typing purposes. Samples were selected from reference receipts to represent the diversity of *Staphylococcus aureus* in England & Wales.

**Table 1** WGS-based MLST results derived from *Salmonella* isolates mixed with other bacteria.

| K-mer identification of primary sample | K-mer identification of secondary sample | ST derived from | |
|---|---|---|---|
| | | WGS-mapping approach (MOST) | WGS-assembly based (BIGSdb) |
| *Proteus mirabilis* WGLW4 | *Salmonella enterica* subsp I *enterica* | 48 | Undetermined |
| *Proteus mirabilis* C05028 | *Salmonella enterica* subsp I *enterica* | 15 | Undetermined |
| *Proteus mirabilis* WGLW4 | *Salmonella enterica* subsp I *enterica* | 19 | 19 |
| *Proteus mirabilis* C05028 | *Salmonella enterica* subsp I *enterica* | 198 | Undetermined |
| *Proteus mirabilis* WGLW4 | *Salmonella enterica* subsp I *enterica* | 897 | 897 |
| *Proteus mirabilis* BB2000 uid214430 | *Salmonella enterica* subsp I *enterica* | 46 | 46 |
| *Klebsiella pneumoniae* subsp. *pneumoniae* KpQ3 | *Salmonella enterica* subsp I *enterica* | 16 | Undetermined |
| *Klebsiella pneumoniae* subsp. *pneumoniae* KpQ3 | *Salmonella enterica* subsp I *enterica* | 414 | 414 |
| *Salmonella enterica* subsp I *enterica* | *Escherichia coli* K 12 substr W3110 uid161931 | Novel allele | 11 |
| *Escherichia coli* K 12 substr W3110 uid161931 | *Salmonella enterica* subsp I *enterica* | 515 | 515 |
| *Proteus mirabilis* C05028 | *Salmonella enterica* subsp I *enterica* | 16 | Undetermined |
| *Proteus mirabilis* C05028 | *Salmonella enterica* subsp I *enterica* | 543 | 543 |
| *Proteus mirabilis* WGLW4 | *Salmonella enterica* subsp I *enterica* | 34 | Undetermined |
| *Proteus mirabilis* WGLW4 | *Salmonella enterica* subsp I *enterica* | 34 | Undetermined |

C. GBRU receives isolates of *Campylobacter* from hospital diagnostic microbiology laboratories and Food Water and Environmental laboratories from England & Wales. From these, representatives of many different STs were included.

***Salmonella* isolates, including those mixed with other species and isolates with low coverage***

Isolates submitted to the above reference laboratories are most often pure cultures, but a small proportion of samples do contain a mixture of organisms. Kmer ID software (https://github.com/phe-bioinformatics/kmerid) was used to identify samples containing mixed species. Samples containing *Salmonella* mixed with other species were used to test the sensitivity of WGS-based MLST methods (Table 1).

Samples with lower than the expected coverage of genomic data can be revealed from the "minimum consensus depth" value. To test how sensitive WGS-based MLST methods were when processing low coverage samples we used *Salmonella* samples with minimum read depth values of 1–10 (Table S1).

### Isolates mixed *in-vitro*
#### Intra-species mixed samples (Strepococcus pneumoniae)

In order to determine how sensitive WGS-based MLST methods are when processing intra-species mixed samples, we assembled artificial mixes of different *S. pneumonaie* types from previously extracted genomic DNA, at different ratios (Table 2).

### DNA extraction and assembly of artificial mix *S. pneumonaie*

DNA was extracted from *Campylobacter* sp., *Salmonella* sp., *Staphylococcus aureus* and *Streptococcus pneumoniae* samples via Qiasymphony (Qiagen GmBH, Hilden, Germany) and quantified (Glomax, Promega, Madison, WI, USA).

**Table 2** WGS-based MLST results derived from DNA of different *S. pneumonaie* types mixed in different ratios.

| S. pneumonaie types and ratio of DNA mixes | Max percentage non-consensus base values derived from MOST software | ST derived from | |
|---|---|---|---|
| | | WGS-mapping approach (MOST) | WGS-assembly based (BIGSdb) |
| 90% ST 4149: 10% ST 5006 | 17.2 | 4149 | Undetermined |
| 80% ST 4149: 20% ST 5006 | 31.0 | 4149 | 4149 |
| 70 % ST 4149: 30% ST 5006 | 40.5 | 4149 | Undetermined |
| 60% ST 4149: 40% ST 5006 | 49.4 | 4149 | Undetermined |
| 50 % ST 4149: 50% ST 5006 | 50.3 | Novel allele | Undetermined |
| 75% ST 1012: 25% ST 2865 | 37.9 | 1012 | Undetermined |
| 50% ST 1012: 50% ST 2865 | 48.2 | Novel allele | Undetermined |
| 75% ST 7181: 25% ST 7219 | 31.7 | 7181 | 7181 |
| 50% ST 7181: 50% ST 7219 | 47.4 | 7219 | 7219 |
| 50% ST 7219: 25% ST 2865: 25% ST 5316 | 49.6 | Novel allele | Undetermined |
| 50% ST 5316: 50% ST 574 | 49.4 | Novel allele | Undetermined |
| 25% ST 5316: 25% ST 123: 25% ST 7219: 25% ST 574 | 46.7 | *NOVEL ST. (no SLV) | Undetermined |

In order to make up intentionally mixed *S.pneumoniae* samples for WGS, DNA extracted from *S.pneumoniae* isolates were mixed at different ratio to give a mixed concentration of 25 ng/µl in a final volume of 75 µl (Table 2). The DNA from isolates:

1. ST 5006 and ST 4149 were mixed in the ratios-10%:90%, 20%:80%, 30%:70%, 40%:60% and 50%:50%
2. ST 2865 and ST 1012 were mixed in the ratios-25%:75% and 50%:50%
3. ST 7219 and ST 7181 were mixed in the ratios-25%:75% and 50%:50%
4. ST 5316 and ST 574 were mixed in the ratios-50%:50%
5. ST 2865, ST 5316 and ST 7219 were mixed in the ratios-25%:25%:50%
6. ST 123, ST 574, ST 5316 and ST 7219 were mixed in the ratios-25%:25%:25%:25%.

## WGS, quality assessment and species identification

Samples for WGS sequencing were submitted to the Genomic Sequencing Unit at PHE. Illumina Nextera DNA libraries were constructed and sequenced using the Illumina HiSeq 2500. Afterwards, the samples were deplexed using the Casava 1.8.2 (Illumina inc., San Diego, CA, USA) and the FASTQ reads were quality trimmed using Trimmomatic (*Bolger, Lohse & Usadel, 2014*) to remove bases with a quality PHRED score below 30 from both ends. K-mer ID software was used to compare the sequence reads with a panel of curated NCBI Refseq genomes to identify the species.

## MLST determination

To extract MLST from *Campylobacter* sp., *Staphylococcus aureu*, *Streptococcus pneumoniae and Salmonella* sp.,the respective allele and ST/profile definitions were downloaded from

**Table 3** MLST results derived using conventional method and WGS.

| Workflow names | Number of samples | Total number of full MLST results derived from | | |
|---|---|---|---|---|
| | | WGS-mapping approach (MOST) | WGS-assembly based (BIGSdb) | Conventional method |
| **Isolates in pure culture** | | | | |
| *Campylobacter Sp.* | 120 | 119 | 112 | 99 |
| *Streptococcus pneumoniae* | 98 | 98 | 98 | 96 |
| *Staphylococcus aureus* | 105 | 105 | 105 | 105 |
| **'Difficult' samples (mixed cultures and those with low coverage)** | | | | |
| Intra species *Streptococcus pneumoniae* | 12 | 7 | 3 | nt[a] |
| Mixed *Salmonella sp* with other bacterial species | 14 | 13 | 7 | nt[a] |
| Low coverage genomic *salmonella* data | 29 | 29 | 27 | nt[a] |

**Notes.**
[a]nt indicates samples not tested.

http://pubmlst.org/data/ and http://mlst.warwick.ac.uk/mlst/dbs/Senterica/Downloads_HTML in August 2015.

STs were determined using:

1. Conventional and WGS based MLST methods from pure isolates in order to compare the conventional method against WGS based MLST.
2. Only WGS based MLST methods from a set of intra and inter species mixed samples and those with low coverage genomic data to investigate the sensitivity of WGS based MLST methods.

The numbers of samples tested via each method are shown in Table 3.

## MLST via conventional sequencing

Alleles were initially amplified by PCR and DNA sequenced using Sanger sequencing. Sanger sequencing was carried out using Applied Biosystems 3720X DNA analyser. Bionumerics version 6.1 was then used to determine the alleles and ST. Bionumerics assigned an allele if the assembled reads matched 100% to the locus variant sequence with zero SNP/INDELs using BLAST. STs were determined using this methodology from set of pure isolates (*Campylobacter* sp., *Staphylococcus aureus* and *Streptococcus pneumoniae* samples).

## MLST via WGS based mapping

At the time that this validation study took place the only available mapping-based approach was SRST (version 1) (*Inouye et al., 2012*). Following initial testing, SRST was modified and the resulting software called "Metric Oriented Sequence Typer" (MOST). Bowtie2 was chosen as the global aligner (rather than BWA) due to the greater sensitivity that we have observed with Bowtie2. MOST uses the output from the Bowtie2 mapping to report percentage coverage across the allele length and the "maximum percentage of non-consensus bases" at any position. The latter value enables the user to identify potentially mixed samples and stop mixed samples not to be reported as novel allele.

Tewolde et al. (2016), *PeerJ*, DOI 10.7717/peerj.2308

The "Max percentage non-consensus bases" value is calculated for each position by using the following formula:

Percentage non-consensus bases = (Number of reads mapped to reference sequence with non-consensus base/Total number of reads aligned to reference sequence) $*$ 100.

Once the percentage non-consensus bases are calculated, the maximum percentage non-consensus base value is determined and reported.

Finally, MOST was adjusted to infer *Salmonella* serotype from the ST value using a PHE *Salmonella* serotype database (*Ashton et al., 2016*). For a full list of other modifications please refer to Supplemental Information 1. MOST is available as open-source software (https://github.com/phe-bioinformatics/MOST).

In addition to the samples used for the conventional ST methodology, STs were also determined from samples with intra and inter species mixes and those with low coverage from the genomic data in order to determine the sensitivity of MOST.

Also included in Table S1 is the time taken to derive a 7 locus MLST result using MOST.

### MLST using BIGSdb—a WGS-assembly based approach

Sequence reads from the same samples described in the previous section were assembled using Spades (version 2.5.1) *de novo* assembly software with the following parameters 'spades.py–careful-1 strain.1.fastq.gz-2 strain.2.fastq-t 2-k 21,33,55,77'. The resulting contigs were uploaded to PubMLST (which runs an instance of BIGSdb) for determination of their STs (*Jolley & Maiden, 2010*).

## RESULTS

### Conventional MLST vs WGS-based MLST—for pure cultures

WGS based MLST yielded via MOST returned full MLST profiles from 99.7% (322) of the 323 isolates tested. This compared to 97.5% (315) via assembly and BIGSdb, and 92.9% (300) by conventional MLST (Table 3). The concordance between samples that return a full MLST profile by conventional MLST and both WGS methods was 100% (Table 3). For *21 Campylobacter* sp and 2 *Streptococcus pneumoniae* samples, a full MLST profile was not returned via the conventional method due to poor sequence quality.

### WGS-mapping based MLST vs WGS-assembly based MLST

Having established the superiority of WGS based MLST over conventional MLST for sensitivity of ST determination from pure cultures, we investigated the accuracy (including the assessment of quality) of different WGS analyses for samples with low coverage and for samples with more than one organism.

From 29 samples that yielded low coverage *Salmonella* genomic data, the WGS-mapping approach (MOST) and WGS-assembly approach (assembly and BIGSdb) returned 100% (29) and 93.1% (27) full MLST profiles, respectively (Table 3). The WGS-assembly based approach did not return full profiles for 2 samples due to truncation of a contig that contained a MLST locus and for the other sample BIGSdb returned two variant matches for the *thrA* allele. BIGSdb identified two variant matches for *thrA* (*thrA67* and *thrA489*) alleles due to the present of duplicated allele within the BIGSdb database. *thrA489*

was removed from curator's database (http://mlst.warwick.ac.uk/mlst/dbs/Senterica/Downloads_HTML) but this was not addressed in BIGSdb at the time.

From 14 *Salmonella* isolates mixed with other bacterial species (Table 1), the WGS-mapping approach (MOST) returned 92.9% (13/14) full MLST profiles whereas the WGS-assembly approach returned full profiles for only half of the samples (50% or 7/14) (Table 3). The WGS-assembly approach did not return full profiles for 7 samples. Three of these were due to contigs that were truncated in a target region (MLST allele), a further three returned two *thrA* allele variants via BIGSdb. The remaining sample had an 'N' introduced in the *aroC* allele. The WGS-mapping approach (MOST) reported ambiguous result (Novel) for 1 sample with "max percentage non-consensus bases" of 50%. This indicates that the sample is mixed at ratio around 50:50. Samples that are mixed might be assigned a novel allele due to the fact the percentage of mixed bases at each position can vary according due to stochastic processes during library preparation and sequencing. For example at one position a mixed base can be 52:48 and at another position the ratio may be 48:52 and therefore it is possible to link allelic calls at each position across the entire sequence with high confidence.

The "max percentage non-consensus bases" quality metric is useful to identify mixed samples and also stops mixed samples to be reported as Novel. The WGS-assembly approach returned 11 allele matches and an ST.

From 12 samples constructed *in vitro* to contain more than one ST of *Streptococcus pneumoniae* we found the WGS-mapping approach (MOST) returned the expected MLST results for 58% (7/12), whilst the WGS-assembly approach (via BIGSdb) returned the expected MLST results for only 25% (3/12) of samples (Table 3). Thus, the mapping based software, MOST, was more sensitive than the assembly based approach. Of the four samples that returned full profiles via MOST, three returned two allele variants using BIGSdb. It reported the following alleles: *ddl* (ddl14 and ddl339) and *spi* (spi2 and spi17) rather than the correct designations: *ddl* (ddl1 and ddl339) and *spi* (spi2 and spi6). The fourth sample has a contig that was truncated in the *gki* allele region. For four *S.pneumoniae* isolates that were mixed at 50%:50% ratio, both WGS based methods did not return correct profile (Table 2).

## MOST quality metrics accurately informed mixed and low coverage samples

Unlike assembly based approaches, the MOST mapping based approach provided a "minimum consensus depth" quality metric that informed low coverage, as well as the "max percentage non-consensus base value" which was informative for identifying mixed samples.

For the 29 samples that yielded low coverage *Salmonella* genomic data, the "minimum consensus depth" values reported by MOST did demonstrate that the samples have low sequence depth (Table S1).

For the mixed samples containing more than one ST of *S. pneumoniae* the "max percentage non-consensus base" values reported by MOST demonstrated the presence of a mixture but also returned the ST of the majority strain within the mixture. However the ratios of the mixtures detected by MOST were consistently higher than the ratios provided by the laboratory. For example samples mixed at ratio 50:50%, 40:60%, 30:70%, 20:80%,

10:90% gave "max percentage non consensus base" values of 50%, 49%, 40%, 31% and 17%, respectively (Table 2), and probably reflects a combination of laboratory pipetting inaccuracies during construction of the mixes, and randomness in the distribution of reads across the MLST loci.

## DISCUSSION

This study revised, tested and validated mapping-based and assembly-based software whose purpose was to extract STs from short-read WGS data by comparing the results with those from the conventional (PCR amplification and Sanger sequencing) MLST methods. Having established the superiority of WGS based methods, we then went on to compare the performance of two WGS data analysis approaches (assembly and mapping) to determine their accuracy against samples that contained more than one organism and low coverage data.

The superiority of WGS based methods was evidenced by the greater number of full MLST profiles as compared to the conventional method. Additional evidence was provided by the complete concordance between the results of conventional and WGS based methods, as well as no instances where only the conventional method returned a full MLST profile. Between the two WGS approaches our comparison indicated that MOST returned, 5% (19) more full MLST profiles than an assembly based approach (Table 3). MOST was particularly effective when handling data from samples with intra- and inter-species mixes. Moreover the quality metric values that it assigns flag mixtures such as these as well as low coverage data. In this respect as well determining the ST from pure samples, it is also suitable for determining the ST from a contaminated or impure sample. The importance of this benefit in the environment of a routine microbiology laboratory cannot be understated, for example we found that 1.5% ($n = 335$) of the cultures of *Salmonella* referred for typing were mixed with other species and 4.9% ($n = 1,060$) contained more than one strain. PHE National Infections Service reference laboratories have selected and used MOST to extract the MLST profile as part of its bioinformatics pipelines. To date (18th March 2016), our reference laboratories have extracted MLST data from over 37,000 samples (21,237 *Salmonella*, 4,256 *Streptococcus pyogenes*, 1,579 *Campylobacter*, 2,920 *Streptococcus pneumoniae*, 3,936 *Escherichia coli*, 1,887 *Staphylococcus aureus*, 1,200 *Listeria monocytogenes* and 700 *Streptococcus agalactiae*) via MOST.

As part of the MOST development we included additional utility to infer serotypes for *Salmonella*. This functionality inferred the serotype from the MLST profile based on a database of previously determined conventional serotyping results and showed 96% ($n = 6,616$) concordance between the MOST and conventional results (*Ashton et al., 2016*). Six months after our implementation of MOST an updated version of SRST (version 2) was released (*Inouye et al., 2014*). Whilst this update included the addition of local mapping alignment, it did not include the additional database analysis component we used for inferring serotype, otherwise our tests indicated agreement with MOST results, except for one sample for which SRSTv2 returned a different type to the conventional type (Table S1).

### Funding

The authors received no funding for this work.

### Competing Interests

The authors declare there are no competing interests.

### Author Contributions

- Rediat Tewolde conceived and designed the experiments, performed the experiments, analyzed the data, contributed reagents/materials/analysis tools, wrote the paper, prepared figures and/or tables, reviewed drafts of the paper.
- Timothy Dallman conceived and designed the experiments, analyzed the data, contributed reagents/materials/analysis tools, reviewed drafts of the paper.
- Ulf Schaefer analyzed the data, contributed reagents/materials/analysis tools, reviewed drafts of the paper.
- Carmen L. Sheppard, Bruno Pichon and Craig Swift performed the experiments, analyzed the data, contributed reagents/materials/analysis tools, prepared figures and/or tables, reviewed drafts of the paper.
- Philip Ashton analyzed the data, contributed reagents/materials/analysis tools, prepared figures and/or tables, reviewed drafts of the paper.
- Matthew Ellington wrote the paper, reviewed drafts of the paper.
- Jonathan Green conceived and designed the experiments, reviewed drafts of the paper.
- Anthony Underwood conceived and designed the experiments, performed the experiments, analyzed the data, contributed reagents/materials/analysis tools, wrote the paper, reviewed drafts of the paper.

### Data Availability

  GitHub: https://github.com/phe-bioinformatics/MOST.

### Supplemental Information

Supplemental information for this article can be found online at http://dx.doi.org/10.7717/peerj.2308#supplemental-information.

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
