# Peer review of "MOST: a modified MLST typing tool based on short read sequencing"

_PeerJ, doi:10.7717/peerj.2308_

## Round 0.1 · original submission · Minor Revisions

Please aim to address all the reviewers' comments and questions in your rebuttal and resubmission.

·

Basic reporting

The mixed datasets used to test the program have not been made available. These would be useful for testing and comparison with other tools. These should be uploaded to the ENA and accession numbers provided in the supplementary table.

The software itself is available and published with an open-source license.

Experimental design

The experimental design appears robust.

Validity of the findings

The findings appear to be valid.

Additional comments

This paper describes a modified tool to extract MLST designations from short read sequencing data using mapping. The manuscript is clearly written and adequately describes the software, reassuringly showing full concordance between Sanger sequencing and WGS using both assembly and mapping. The mapping approach used by the software was shown to be more sensitive than assembly, especially with poor quality samples. The quality metrics that can be produced for the mapping approach are clearly important for use within public health and clinical settings. All my comments are minor:

Lines 156-157: The databases themselves were not downloaded. Allele and ST/profile definitions were downloaded.

Lines 186-187: While correct, since multiplication and division have equal precedence and are read left to right, use of parentheses in the formula would make it clearer, i.e. (number of reads mapped to ref seq with non-consensus base / number of reads aligned to ref) * 100.

Line 204: It is not clear where the contigs were uploaded to. BIGSdb is a software platform – was this run locally, or were the contigs uploaded to PubMLST (which runs instances of BIGSdb)? BIGSdb is not cited in the references.

Lines 230-232: If BIGSdb identified two variant matches for a thrA allele then these variants must both have been present in the assembly. Was this due to a mis-assembly, gene duplication or sample contamination?

An indication of the time required to process the samples would be useful. Clearly for 7-locus MLST, the assembly process itself will be the bottleneck in the assembly-based method, and I would imagine a read mapping approach to be quicker, but with the advent of cgMLST schemes where you may wish to extract 2000 loci, how well does the method scale?

Finally note to editorial/production: Lines 111, 194: Hyperlinked URLs to the GitHub archives are not valid in the PDF (although this is because the formatting has inserted the line number in to the hyperlink). This is an issue with the review documents and not with the author supplied files.

·

Basic reporting

The authors described a new pipeline named "MOST", which was modified from SRST (v1) and used a mapping based approach to determine MLST types from WGS data. They also proven that WGS based approaches are more sensitive than the conventional approach.

Experimental design

The authors described most of the methods clearly, with three exceptions.
1/ the MLST references used in the Bowtie2 alignments were not well versioned. New alleles were generated by all MLST sites every day and different sets of references can potentially change the results of MOST. Well versioned databases are important for not only the reproductions of this study, but also the comparisons of MLST typing results generated from different institutes and/or different time points in the same institute.

2/ MOST used "max percentage non-consensus bases" and "minimum consensus depth" to describe the qualities of allele callings. However, there was no validation about the correlations between the reported qualities and the correctness of callings. For example, 5/8 of the mixed S. pneumonaie samples that had >40% of max non-consensus bases were not assigned to correct ST types.

3/ One important process in MLST typing is the identification of novel alleles. It is questionable that whether a mapping based approach can find new allele sequences efficiently. In particular, there were two cases that were known to be problematic for mapping based methods:
A) Alleles that have novel INDELs within the last several bases at the edges of a locus.
B) Novel alleles that were >5% divergent from any known alleles.

Validity of the findings

There are several discrepancies between the reported summaries in the main text and the supplementary table.
1. Line 38, "Of the 325 samples, ". There was at least one duplication of "131426_streptococcus-pneumoniae" and "131426_tttstreptococcus-pneumoniae" in the supplementary table.

2. Line 38, "(n=316)". There were 317 MLST profiles derived from assembly-based approach in the supplementary table.

3. Line 40, "The concordance between samples that were typed by conventional (92.9%) and both WGS methods was 100%". There were two cases ("131399_streptococcus-pneumoniae" and "131426_streptococcus-pneumoniae") that the conventional approach gives new alleles whereas the WGS-based approaches gave a known ST number.

4. There is one sample ("42018_Cjejuni") in the supplementary table that has a max of 97% non-consensus bases, which should never go above 50%.

Additional comments

There are several other minor points:
1/ Over half of the failed callings in bigsDB were due to the presences of two variants in one of the loci. Does one of these two variants match with the allele derived from MOST? The bigsDB platform is designed to define correct alleles more than complete ST profiles and tends to be over-sensitive when there were multiple paralogs in the assembly.

2/ Table 1. Only species names can be in italic.

3/ Table 2. There were only 11 mixes in this table, rather than 12 mixes described in the main text.

4/ Line 262-4. "“max percentage non consensus base” values of 50%, 49%, 40%,31% and 17%, respectively (Table 2), and may reflect laboratory (pipetting) bias during construction of the mixes".

The interprets were not correct. The proportions of mapped reads from mixed strains in each sites of the alignment were randomly distributed. The "max percentage non consensus base" value does not represent the average values of the proportions and is only a biased indicator.

---

## Round 0.2 · accepted · Accept

Thank you for your revised manuscript addressing each of the reviewer's comments. I am satisfied that you have dealt with the outstanding issues that were raised by the reviewers and therefore I am delighted to be able to accept the article.